# Effects of Ultra-Short Pulsed Electric Field Exposure on Glioblastoma Cells

**DOI:** 10.3390/ijms23063001

**Published:** 2022-03-10

**Authors:** Arianna Casciati, Mirella Tanori, Isabella Gianlorenzi, Elena Rampazzo, Luca Persano, Giampietro Viola, Alice Cani, Silvia Bresolin, Carmela Marino, Mariateresa Mancuso, Caterina Merla

**Affiliations:** 1Italian National Agency for Energy New Technologies and Sustainable Economic Development (ENEA), Division of Health Protection Technologies, Via Anguillarese 301, 00123 Rome, Italy; arianna.casciati@enea.it (A.C.); mirella.tanori@enea.it (M.T.); carmela.marino@enea.it (C.M.); 2Department of Ecological and Biological Sciences, University of Tuscia, Largo dell’Università, snc, 01100 Viterbo, Italy; isabella1693@hotmail.it; 3Department of Women’s and Children’s Health (SDB), University of Padova, via Giustiniani 3, 35128 Padova, Italy; elena.rampazzo@unipd.it (E.R.); luca.persano@unipd.it (L.P.); giampietro.viola.1@unipd.it (G.V.); cnalca@unife.it (A.C.); silvia.bresolin@unipd.it (S.B.); 4Division of Pediatric Hematology, Oncology and Hematopoietic Cell & Gene Therapy, Pediatric Research Institute (IRP), Corso Stati Uniti 4, 35127 Padova, Italy

**Keywords:** cancer stem cells (CSCs), glioma, neurospheres, pulsed electric field, transcriptomic analysis

## Abstract

Glioblastoma multiforme (GBM) is the most common brain cancer in adults. GBM starts from a small fraction of poorly differentiated and aggressive cancer stem cells (CSCs) responsible for aberrant proliferation and invasion. Due to extreme tumor heterogeneity, actual therapies provide poor positive outcomes, and cancers usually recur. Therefore, alternative approaches, possibly targeting CSCs, are necessary against GBM. Among emerging therapies, high intensity ultra-short pulsed electric fields (PEFs) are considered extremely promising and our previous results demonstrated the ability of a specific electric pulse protocol to selectively affect medulloblastoma CSCs preserving normal cells. Here, we tested the same exposure protocol to investigate the response of U87 GBM cells and U87-derived neurospheres. By analyzing different in vitro biological endpoints and taking advantage of transcriptomic and bioinformatics analyses, we found that, independent of CSC content, PEF exposure affected cell proliferation and differentially regulated hypoxia, inflammation and P53/cell cycle checkpoints. PEF exposure also significantly reduced the ability to form new neurospheres and inhibited the invasion potential. Importantly, exclusively in U87 neurospheres, PEF exposure changed the expression of stem-ness/differentiation genes. Our results confirm this physical stimulus as a promising treatment to destabilize GBM, opening up the possibility of developing effective PEF-mediated therapies.

## 1. Introduction

Glioblastoma Multiforme (GBM) is the most lethal brain tumor in adults, with a survival time of 12–15 months after initial diagnosis [1]. The highly resistant nature of GBM is largely due to extreme intra-tumoral heterogeneity [2,3,4,5,6,7,8]. As demonstrated in several cancers, GBM is thought to arise from cancer stem cells (CSCs), representing a small percentage of all cancer cells responsible for tumor development and progression [9,10,11]. These cells, characterized by a complex mixture of genetic alterations [12,13], express high levels of stem markers involved in self-renewal and are intrinsically resistant to standard therapies (such as radiotherapy and chemotherapy), causing tumor spread. CSCs are consequently responsible for tumor relapses [14,15,16,17,18].

Notably, the canonical CSC paradigm has been recently revised, and it is now widely accepted that the CSC phenotype is characterized as a dynamic cellular state rather than a fixed population. CSC phenotypic heterogeneity arises from reversible cell state transitions in response to varying intracellular/extracellular signals [19]. Lineage tracing experiments in many solid tumors clearly demonstrated the continuous turnover of CSCs that under epithelial mesenchymal transition can convert from non-CSCs [20,21,22]. The dynamic and bidirectional transition states among epithelial, mesenchymal and one or more hybrid epithelial/mesenchymal phenotypes [23], called “epithelial mesenchymal plasticity,” show a dynamic reversible nature in cancer cells and have a strong impact on cancer progression [19].

In light of this complex scenario, new therapies to treat highly aggressive and heterogeneous GMB are required and remain a main challenge in current oncologic research. Recently, ultra-short pulsed electric fields (PEF) of high amplitude (from few to hundreds of kV/m) and short duration (from a few milliseconds to few microseconds) have emerged as a very powerful physical agent provoking cell reversible or irreversible cell electropermeabilization, both tested in anti-cancer therapies [24,25,26,27]. Indeed, electrically mediated treatments using PEF are rapidly increasing through selective tumor cells targeting using a non-thermal process, resulting in the reversible or irreversible destabilization of cell plasma membranes. In the reversible case, cell membrane permeabilization allows the entrance of toxic drugs (i.e., electrochemotherapy). In irreversible applications, electric pulses disrupt cell homeostasis and consequently cause cell death (irreversible electroporation IRE, or the more recent high-frequency irreversible electroporation H-FIRE approach) [25,28,29,30]. The non-thermal nature of these electric signals preserves the healthy tissues surrounding the tumors, therefore enabling the treatment of localized masses, including in delicate anatomical locations.

The impact of PEF was recently investigated on GBM cell lines, suggesting these electric stimuli are an extremely promising glioma therapy [12]. It also demonstrated that the use of PEF as a novel GBM therapy is related to the fact that it facilitates temporary blood–brain barrier disruption, allowing easy and effective penetration of therapeutic reagents [31,32], as well as the induction of immunogenic cell death for prolonged antitumor immunity [33]. In addition to these important aspects, due to GBM heterogeneity, the characterization of GBM CSCs’ response to these electric pulses represents a new and poorly investigated concept in search of new GBM therapies.

In this context, we demonstrated that a specific signal of this class, named PEF-5 (i.e., 5 electric pulses lasting 40 µs, pulse amplitude 0.3 MV/m, repeated at 1 Hz), irreversibly affected medulloblastoma (MB) CSCs, inducing PEF-surviving cells to better respond to a subsequent exposure to ionizing radiation (IR) [34].

According to the dynamic nature of CSCs, we evaluated the effects of PEF-5 exposure on U87-MG cells maintained in standard culture conditions (U87 ML) or as neurospheres (U87 NS). Our data, obtained by analyzing different endpoints (i.e., cell viability, cell permeabilization, reactive oxygen species (ROS), full transcriptomic analysis and mRNA expression of different genes, as well as clonogenic and invasion potential), showed that PEF-5 exposure substantially influenced the fate of GBM CSCs, differentially regulating many genes involved in hypoxia, inflammation and *P53*/cell cycle checkpoints, with consequent reduction of the cell’s ability to form new neurospheres and to transmigrate in vitro. Due to the well-recognized radio-resistance reported for GBM, in this work, we assessed the PEF-5 capability to affect GBM cells in combination with IR.

## 2. Results

### 2.1. Expression Pattern of Stemness/Differentiation Genes in U87 ML and U87 NS Cells

First, we explored the potential differences in gene signature by looking at the stemness/differentiation pathways in U87 ML (Figure 1a) and U87 NS (Figure 1b) and evaluating, by qPCR, the expression levels of *CD133*, *CD15* and *β-III TUBULIN*. We found a significantly higher level of stemness markers in U87 NS cells compared to U87 ML cells. *CD133* and *CD15* expression levels were 2.8- and 33.1-fold higher in U87 NS versus U87 ML cells, respectively (Figure 1c,d). On the contrary, U87 NS showed a statistically significant lower level of *β-III TUBILIN* (Figure 1e). These results confirmed the dynamic phenotypic state of GBM cells that can modulate their phenotypic features according to specific cell culture conditions.

### 2.2. Membrane Permeabilization and ROS Production after PEF-5 Exposure

To verify the efficacy of PEF-5 to induce cell membrane permeabilization in U87 ML and U87 NS, the uptake of the fluorescent Yo-Pro-1 dye was evaluated by flow cytometry. Graphs in Figure 2a show the percentage of permeabilized cells immediately (T = 0 h) and 3 h (T = 3 h) after exposure. We found 91% of permeabilized cells immediately after exposure in U87 ML compared to 28% in U87-NS cells. This permeabilization phenomenon is transient, as at T = 3 h the dye uptake markedly decreased in both culture conditions (Figure 2a). Nonetheless, U87 NS cells, richer in CSCs, were more resistant to permeabilization. Following cell permeabilization, ROS levels were evaluated by DHE uptake at T = 0 h or at T = 3 h after PEF-5 exposure (Figure 2b). In U87 ML cells, a statistically significant ROS increase was observed after electric pulse exposure exclusively 3 h after exposure. On the contrary, in exposed U87 NS cells, ROS levels were significantly higher at T = 0 h, remaining stable at T = 3 h post-exposure. These results highlighted an overproduction of ROS that clearly indicated cellular stress in response to this exposure.

### 2.3. Survival, Proliferation and Cell Cycle Perturbation

By trypan blue assay, we assessed the cell viability in both U87 ML and NS after PEF-5 exposure to verify the level of cell survival following reversible cell permeabilization (Figure 2a,b). As shown in Figure 2c,e, U87 ML and NS cells reported a similar (22% and 24%) statistically significant decrease in cell viability 24 h after exposure to PEF-5, suggesting that PEF-5-mediated cell perturbations (i.e., transient permeabilization and ROS generation) did not induce massive death in GBM cells, regardless of their CSC content. Furthermore, as shown in Figure 2d,f, PEF-5 exposure induced a 2.5- or 4.7-fold lower proliferation rate in U87 ML surviving cells compared to the 1.3- and 2.0-fold decrease observed in U87 NS at 24 and 48 h after exposure, respectively.

In order to more deeply understand the reason for changes in cell proliferation, we analyzed the cell cycle (Figure 2g–j). The exposure of U87 ML cells did not induce differences in any phase of the cell cycle compared to the sham-exposed cells (Figure 2g,i). Instead, in the enriched CSC population, U87 NS, the exposure to PEF-5 induced a statistically significant increase of G0/G1 with a significantly high decrease in the G2/M phases (Figure 2h,j). In U87 ML cells, the proliferation reduction may be due to a generalized slowdown of all phases of the cell cycle; on the contrary, in the heterogeneous U87 NS cells, there is an imbalance in the different phases of the cycle and an accumulation of cells in G1.

### 2.4. Transcriptomic Profiles of U87 ML and NS Cells and Validation of mRNA Expression

#### 2.4.1. Affymetrix Gene Chip Analyses

Following the observed PEF-5 effects in terms of ROS generation, survival and cell cycle changes, to more widely screen significantly deregulated genes, the gene profile of both U87 ML and NS was examined by Clarion S Affymetrix Gene Chip analyses, after 24 h from PEF-5 exposure. Genes were considered differentially expressed by Significance Analysis of Microarray (SAM) with FDR < 0.05. Relative heat maps with hierarchical clustering generated by normalization of CEL files (detailed in Section 4.5), derived from whole transcriptional analysis, are shown in Figure 3a. The comparison between sham-exposed and PEF-5-exposed U87 ML cells allowed the identification of 156 differentially expressed transcription clusters (TCs; 115 upregulated and 41 downregulated TCs). In the same way, analysis of RNA derived from U87 NS cells disclosed 60 TCs as differentially expressed between sham and exposed cells (39 upregulated and 21 downregulated; Figure 3a,b). As shown by the Venn diagram in Figure 3b, fewer genes were modulated in U87 NS compared to ML cells (60 vs. 156 TCs observed in U87 ML). Of note, only a few commonly deregulated genes were identified after PEF-5 exposure. In particular, three upregulated genes, i.e., *Hyaluron synthase 2* (*HSA2*), *proline rich protein 2A* (*SPRR2A*) and *2D* (*SPRR2D*), were the same as U87 ML and NS cells (box in Figure 3b). These genes have been described as components of the plasmatic membrane, having a role in membrane integrity, cell survival and inflammatory responses even post radiation and stressful environmental conditions [34,35,36,37].

In order to identify the potential pathways and intracellular signaling affected by pulse exposure, the gene lists have been analyzed for their enrichment in pathways reported in multiple MSigDB gene sets, including Hallmarks and C2cp (common pathways). In particular, U87 ML cells displayed significant overactivation of multiple hypoxia-dependent targets, with biological processes involved in the response to intracellular oxidation, positive perturbation of P53/cell cycle and alterations of NFkB and TNFα-dependent inflammation signaling (Figure 3c). On the contrary, PEF-5 exposure negatively affected different processes involved in cellular remodeling and in regulation of the mTOR signaling pathway, known to be involved in tumor metabolism [38]. Moreover, U87 ML PEF-5-treated cells showed a clear enrichment of apoptosis and interleukin and cytokine-induced senescence secretory phenotype [39] (Figure 3c,e). Analysis of differentially expressed genes in U87 NS cells displayed a similar modulation of genes involved in hypoxia, intracellular oxidation, inflammation and in P53/cell cycle deregulation (Figure 3d). In particular, *interleukin 6* (*IL-6*) with a significant alteration of NFkB/TNFα-dependent inflammation pathways may suggest a common signature between cell types. Moreover, in U87 NS PEF-5-exposed cells, a significant enriched deregulation of DNA repair processes and the upregulation of gene transcription and RNA processing seems to occur (Figure 3e).

#### 2.4.2. qPCR Validation of Transcriptome Data for Selected Genes

With the aim of validating the transcriptomic data, we evaluated the expression levels of genes involved in the most deregulated pathways 24 h after PEF-5 exposure i.e., cell cycle (Figure 4a,b) and hypoxia-inflammation signaling (Figure 4c,d). In Figure 4a,b, we observed a significant increase of *P53* and *P21*, the damage sensor proteins, in both U87 ML (*P53* = +0.2 fold, *p* = 0.005; *P21* = +0.4 fold, *p* = 0.008) and U87 NS cells (*P53* = +0.4 fold, *p* = 0.003; *P21*= +0.5 fold *p* = 0.002), in agreement with our transcriptomic analysis. Due to the involvement of P53 in cell cycle regulation, we verified whether PEF-5 exposure altered the levels of different cell cycle components that might have contributed to G1 arrest in U87 NS cells. 

The expression levels of the two cyclin-dependent kinases, *CDK2* and *CDK4*, were significantly affected in both cell conditions (*CDK2* −0.6 fold, *p* = 0.02 and −0.5 fold, *p* = 0.05 in U87 ML and NS, respectively; *CDK4*, −0.6 fold, *p* = 0.002; −0.6 fold, *p* = 0.04 in U87 ML and NS, respectively). On the other hand, in U87 ML cells, no significant change was observed in *CYCLIN D1* mRNA level (Figure 4a), while it was upregulated in U87 NS cells (+0.6 fold, *p* = 0.04; Figure 4b). In the same way, we observed an inverse behavior in the *GADD45* levels, a gene involved in cell cycle progression, downregulated in U87 ML cells (−0.5 fold, *p* = 0.009) and upregulated in U87 NS (+0.4 fold, *p* = 0.03; Figure 4a,b). These data demonstrated that PEF-5 exposure induces a complex cell cycle perturbation, possibly enabling a consequent cell-dependent fate alteration. The significant alteration in many genes involved in cell cycle progression (Figure 4b), such as *CYCLIN D1* (+0.6 fold, *p* = 0.04), *GADD45a* (+0.4 fold, *p* = 0.03) and *CDK4* (−0.6 fold, *p* = 0.04), may explain the cell accumulation in the G1/G0 phase and the G2/M phase reduction observed in Figure 2j. Indeed, the complex of *CYCLIN D1* with *P21*, concurrently with a selective decrease of both *CDK2* and *CDK4,* appears responsible for the decrease of cell proliferation and cell cycle arrest (Figure 2f,j). To further validate our transcriptomic analysis, hypoxia/inflammation signaling was analyzed by mRNA expression levels of *interleukin-6* (*IL6*) and *chemokine* (C-C motif) *ligand 20* (*CCL20*), *cyclooxygenase* (*COX*)-*2* and *inducible nitric oxide synthase* (*iNOS*), all downstream genes of NF-KB signaling (Figure 4c,d). Data showed a significant upregulation of *IL6* mRNA (+4.0 fold, *p* = 0.006 and +0.4 fold, *p* = 0.001 in U87 ML and NS, respectively) and *CCL20* (+3.9 fold, *p* = 0.03 and +0.6 fold, *p* = 0.0008 in U87 ML and NS, respectively). Furthermore, *COX-2* (+9.6 fold, *p* = 0.0003 in U87 ML) and *iNOS* (+5.0 fold, *p* = 0.02 and +0.3 fold, *p* = 0.03 in U87 ML and NS, respectively) were significantly overexpressed compared to sham-exposed cells in both cell lines. Of note, the alteration of all genes analyzed was modest in U87 NS cells compared to U87 ML cells.

Furthermore, deregulation of *β-CATENIN* mRNA expression, observed in both U87 ML (−0.5 fold, *p* = 0.03) and U87 NS cells (+0.4 fold, *p* = 0.02), together with *iNOS* upregulation demonstrated the ability of PEF-5 to induce changes in oxygen-sensitive transcriptional regulators, including NF-kB and WNT/β-Catenin [40,41], thus providing support to our evidence of ROS generation after PEF-5 exposure.

#### 2.4.3. Effects of PEF Exposure on the Stemness/Differentiation Status of GBM Cells

To verify if PEF-5 could affect self-renewal and possibly differentiation processes extremely important in CSCs’ fate, we also investigated the gene expression of *NANOG*, *OCT4* and *SOX2* 24 h after the exposure. In U87 ML cells, we found a statistically significant downregulation of *SOX2* (−0.24 fold, *p* = 0.05; Figure 4e). Notably, in U87 NS cells, all genes were upregulated (*NANOG*= +1.24 fold, *p* = 0.0048; *OCT4* and *SOX2*= +0.5 fold, *p* = 0.01; Figure 4f). To evaluate whether PEF-5 could also change the balance between stemness and differentiation in GBM cells, we evaluated the expression of stemness (*CD133* and *CD15*) and differentiation (*β-III TUBULIN*) markers. PEF exposure did not induce perturbation of *CD15*, *CD133* and *β-III TUBULIN* in U87 ML cells compared to their unexposed counterparts (Figure 4e). Notably, exposure to PEF-5 induced a significant decrease in *CD133* (−0.4 fold, *p* = 0.001) and *CD15* (−0.6 fold, *p* = 0.0003) and a significant increase of *β-III TUBULIN* (+ 0.2 fold, *p* = 0.03) in U87 NS cells (Figure 4f), suggesting selective action of PEF exposure toward cells with a more undifferentiated status.

### 2.5. Functional Testing after PEF-5 Exposure

#### 2.5.1. Clonogenic Survival Assay and IR Combined Treatment

To assess the possible influence of PEF-5 on GBM clonogenic capacity, the survival clonogenic assay was performed on both first neurospheres derived from U87 ML (U87 NS1) and second generation neurospheres derived from U87 NS cells (U87 NS2). As shown in Figure 5, exposure to PEF-5 efficiently decreased the clonogenic capacity in both neurosphere generations (about 2-fold lower than sham-exposed cells; *p* < 0.0001 in U87 ML and *p* < 0.001 in U87 NS). These results again suggested a strong ability of PEF-5 to modify the stem phenotype of CSCs, decreasing their self-renewal capacity (Figure 5a–d). As radiosensitivity increases with cell differentiation status, we tested the effectiveness of a combined protocol, irradiating U87 ML and U87 NS cells with increasing IR doses (2, 5, 8 Gy), 3 h after PEF-5 exposure, as suggested in our previous work [34]. As shown in Figure 5a–d, the combined treatment of PEF-5 and IR was able to reduce clone formation as a function of delivered doses, suggesting an additive effect between these two physical agents.

#### 2.5.2. Invasion Assay after PEF-5 Exposure

As invasiveness is another pathophysiological feature of human malignant gliomas, the effects of PEF-5 on the invasiveness and migration of GBM cells were checked in vitro by Transwell Matrigel invasion assay. As shown in Figure 6a–d, the invasiveness of U87 ML and NS cells exposed to PEF-5 was strongly reduced compared with the control groups. U87 ML PEF-5 exposed cells showed a corresponding decrease of the invasion of 34% and U87 NS cells of 47%. This finding indicates the capacity of PEF-5 alone to inhibit the invasive ability of GBM cells in vitro. Additionally, for this assay, a combined protocol, PEF-5 + IR, was carried out using increasing IR doses (2, 5, 8 Gy) delivered 3 h after PEF-5 treatment. No further decrease in invasion ability was observed in only irradiated U87 ML samples (Figure 6c), instead, a decrease of 28% (*p* = 0.006) and of 29% (*p* = 0.009) was observed in U87 NS cells irradiated with 5 and 8 Gy, respectively (Figure 6d). This result may point out a migration response mediated by the cell types and the specific delivered radiation dose. After the combined protocol of exposure, no significant changes were observed in migration and invasion abilities of these two cell types compared to those observed in the PEF-5-only treated groups. This result highlights the radio resistance of GBM cells in both culture conditions. Notably, PEF-5 treatment alone was as effective as the combined treatment with radiations at all the delivered doses in both U87 ML (PEF vs. 2Gy, −41% *p* = 0.04; PEF vs. 5Gy, −32% *p* = 0.0096; PEF vs. 8Gy, −25% *p* = 0.016) and U87 NS cells (PEF vs. 2Gy, −41% *p* = 0.04).

## 3. Discussion

Despite cancer treatment progress with surgery, chemotherapy and radiotherapy, no effective treatments are available for GBM patients and a new and more effective therapeutic strategy that effectively destabilizes GBM CSCs would represent a major development. In the current study, a specific electric pulse protocol (PEF-5), characterized by a high amplitude and a short duration, has proven to significantly affect the growth of U87-GBM cells and GBM tumor-initiating cells. Considering the dynamic nature of CSCs, the U87 GBM cell line was propagated in two different ways in our experimental design: (i) as a classic monolayer, to investigate a relatively more homogenous population, and (ii) in a more heterogeneous condition as NS. These three-dimensionally organized spheroidal structures contain a nucleus of cells with reduced dimensions, more prone to transform into quiescent or dormant cells, surrounded by a mixture of progenitor stem cells and a very small number of differentiated cells. Our data, in line with the literature [42], confirmed that NS are characterized as having a more undifferentiated status compared to the cells from which they derive. However, NS present a high level of plasticity and possess the ability to dynamically switch between CSC and non-CSC states under the influence of specific stimuli [43,44]. Regardless of cell culture modality, PEF-5 mainly affects the cell plasma membrane [25,29,45,46]. PEF-5 exposure gives rise to an increase in the transmembrane potential, which causes structural membrane defects and permeability disorder. However, PEF-5 is unable to generate irreversible membrane permeabilization, similar to that observed in MB cells [34], in both GBM cultures, which resulted in a lower level of cell death, demonstrating the existence of a PEF-5 toxicity threshold value, higher in U87-GBM than in D283-MB cells. This threshold appears to depend on specific cellular characteristics that may play a key role in predicting the cell response to PEF exposure [12].

The significantly lower ability of U87 NS to be permeabilized compared to U87 ML could be related to the different membrane composition/complexity of this more heterogeneous population. The clear enrichment in stem-type transmembrane proteins, such as CD133 and CD15, observed in U87 NS, can determine a different and specific cellular and molecular response to electrical pulse exposure. The lower permeabilization induced in U87 NS by the electric field could also depend on their specific intracellular dielectric characteristics. Indeed, the more differentiated tumor cells (U87 ML) can be identified and separated from the more undifferentiated stem subpopulation (U87 NS) on the basis of specific cytoplasmic redox characteristics [47,48] related to transmembrane proteins [47,48,49].

Although the initial mechanism of PEF-5 action generates only transient electro-permeabilization on U87-GBM cell membranes, the electrical pulses act as a plasma membrane’s stress factors, generating ROS. As second messengers, ROS can regulate many fundamental cell functions, such as growth, proliferation, differentiation and cell death [50,51,52,53]. ROS content may also have profound effects on stem cell fate [54]. The high ROS production observed in U87-GBM cells is independent of culture conditions, albeit with different kinetics, and suggests ROS as one of the key mechanisms in response to PEF-5 in cancer cells. The electrical pulses, generating a perturbation on the membrane through the cytoskeleton [55], may also influence the electrical potential of intracellular organelles, such as mitochondria, anchored to the cytoskeleton itself, causing the consequent ROS increase [56,57]. When ROS production overcomes the cell antioxidant defenses, the subsequent oxidative stress causes negative effects on multiple cell components. The different kinetics of ROS production immediately after exposure to PEF-5 in U87 NS cells compared to U87 ML (only 3 h after exposure) may depend on the different degrees of stemness of the two cell systems. Although CSCs showed a greater activity of ROS buffering systems [54,58], all tumor cells, due to their high metabolism, are more vulnerable to agents that induce even minimal disturbances in ROS production [59]. In both culture systems, the indirect effect of PEF-5, modulating ROS and membrane integrity, induced a significant block in their ability to proliferate, providing a way to selectively inhibit tumor self-renewal capacity.

Therapeutic strategies based on cell membrane modulation, as demonstrated by PEF exposure, could represent an alternative and interesting tool for future GBM treatment. Among the oxidative sensors, we find proteins, such as P53 and P21 [60,61]. Our results showed a statistically significant increase in *P53* (by transcriptomic analysis and mRNA expression) and subsequent induction of *P21* in both U87 ML and U87 NS. This suggests the ability of PEF-5 to propagate damage by signal transduction pathways directly from membranes to the nucleus, generating important perturbations in the cell cycle checkpoint genes [60,61], with consequent inhibition on cell proliferation in both U87 ML and NS cells. Furthermore, in U87 NS, where higher cellular heterogeneity is evidenced, we observed a perturbation in several key checkpoint genes in response to PEF-5 exposure. In particular, the significant increase in *CYCLIN D1* expression, known for its role in regulating cell cycle progression [62], complexing with *P21*, combined with a selective decrease of both *CDK2* and *CDK4,* could explain the accumulation in G1/G0 and the reduction of the G2 phase [63,64] preventing cell cycle progression in U87 NS [65,66]. In the same way, our data showed a significant PEF-5 related downregulation of *CKD2* and *CDK4*, also in U87 ML cells, suggesting that the electric field can potentially act as a powerful class of CDK inhibitors, targeting pivotal players of the cell cycle [64].

Proteins of the GADD45A family, which regulate the balance between repair, cycle arrest, senescence, apoptosis and maintenance of the stem cell pool [67,68], have been linked in recent years to oxidative cell damage [69,70]. In our set-up, the modulation of *GADD45A* expression could, therefore, interfere not only with proliferation and cell cycle perturbations [34,67] but also with deregulation of molecular mechanisms maintaining the balance among stemness, proliferation and differentiation within cancer cells [71,72,73], suggesting the involvement of the ROS-P53-GADD45A axis. Indeed, as reported in the literature, one of the many roles of P53 is to directly bind the NANOG promoter [74] and, through the inhibitory action of P21, it seems to be able to at least partially repress the expression of SOX2 [75], as observed in U87 ML cells. Furthermore, through a complex mechanism of cross-talk, the same proteins NANOG, OCT4 and SOX2, are able to regulate the self-renewal process of CSCs through an action on P53 and ROS-mediated by the interaction with the transcription HIF2a [76,77] and MYC [78]. Thus, future analyses are advisable to fully understand PEF action.

According to other authors [68], in U87 NS cells, the overexpression of *GADD45A* could induce and accelerate the differentiation process through the activation of p38-MAPK [68,79,80]. Therefore, the significant decrease in the expression of stem cell genes, such as *CD133* and *CD15*, in U87 NS suggests that PEF-5 could drive some of the cells present in the neurospheres toward differentiation. In agreement with Boso and collaborators [40], the hypoxia pathway observed by transcriptomic analysis and the deregulation of *β-CATENIN* mRNA expression, observed in both U87 ML NS cells, demonstrated the ability of PEF-5 to induce changes in oxygen-sensitive transcriptional regulators, including NF-kB and WNT/β-catenin [40,41], suggesting a partial induction of neuronal differentiation in exposed cells. This finding is also confirmed by the significant activation of the *β-III TUBULIN* gene expression [40,81]. Although the molecular mechanisms that maintain these processes in CSCs compared to non-CSCs are still unclear, the fine modulation of stemness genes [82], such as important transmembrane proteins, such as CD133 and CD15 [42,83], accompanied by a modulation of cellular checkpoint genes [84,85], appears to be of fundamental importance for the survival of CSCs in maintaining the balance between self-renewal and differentiation [77,86].

The functional data obtained from the analysis of the clonogenic test in vitro confirm a specific effect of PEF-5 treatment on U87 tumor cells. Exposure to PEF-5 induces a significant reduction of clonogenic potential in both U87 NS1 and U87 NS2, inhibiting tumor-initiating cell growth and suggesting a strong capacity of PEF-5 to modify the stem phenotype of U87 CSCs [87]. Combined PEF-5 exposure and IR showed an additive effect of the two physical stimuli with a dose-dependent effect of IR on the cell clonogenic potential.

Notably, cell invasiveness was suppressed in both U87 ML and NS exposed cells. Of note, PEF-5 treatment alone was significantly more effective in decreasing cell migration and invasion than radiation exposure in both U87 ML and NS cells independently of the delivered dose. A possible explanation of this radio resistance observed for cell invasion may come from transcriptomic analysis where the activation of hypoxia and NF-kB pathways, validated by the *COX2*, *CCL20* and *iNOS* downstream genes, may induce a radio-resistance mechanism after PEF-5 exposure [88].

Looking globally at our results, a suitable strategy to increase U87-GBM response to PEF would be to modify the electric pulse parameters in order to induce irreversible damage to cells as in MB CSCs [34], for example in terms of pulse number, amplitude and pulse repetition frequency, to generate signal transduction mechanisms involving stronger phenotypic cell modifications. Another interesting point would be to fully understand the main PEF-5 molecular bio-targets, which will be helpful in guiding the optimal choice of electric protocols. In our opinion, the different cell membrane composition in terms of transmembrane proteins, receptors and associated molecules could be relevant for the observed selective PEF action and further experiments are required to fully elucidate these complex biological and biophysical interactions on other GBM cell lines. Experiments on 3D spheroids as an in vivo-like model of tumors [89] can be further used in the future to assess PEF exposure as an effective therapy on GBM or on other tumor types before moving to in vivo studies.

## 4. Materials and Methods

### 4.1. Cell Culture

The human U87-MG cell line was purchased from American Type Culture Collection (Manassas, VA, USA) and was maintained in Dulbecco’s Modified Eagle’s Medium–high glucose (DMEM) containing 10% (*v*/*v*) fetal bovine serum (FBS), 100 units/mL penicillin and 100 mg/mL streptomycin. Cells were incubated at 37 °C with 95% air and 5% CO_2_.

To enrich U87 cells in CSC content (neurosphere assay), they were cultured in a defined serum-free neural stem cell medium: 10 ng/mL of insulin, 2 mg/mL of heparin, 9.6 ng/mL of putrescine, 0.063 ng/mL of progesterone, hepes 5 nM, B27 1X without vitamin A, 20 ng/mL basic fibroblast growth factor (h-FGF) and 20 ng/mL of epidermal growth factor (h-EGF); provided by Sigma Aldrich, Burlington, MA, USA.

### 4.2. Cell Exposure to PEF-5 and Ionizing Radiation IR

Cells were exposed to a specific and well-characterized electric field (5 electrical pulses with an inter-pulse interval of 1 Hz), as previously described [34]. Briefly, all experiments were performed using electroporation cuvettes (0.1 cm gap, Bio-Rad, Italy) connected to a pulse generator (Schaffner NSG504, Eichenau, Germany). The delivered electric pulses have an exponential shape; their amplitude is maintained constant at 300 V (for an electric field of 0.3 MV/m) and the pulse duration is 40 μs at full width at half maximum. This exposure protocol is indicated as PEF-5. The pulse waveforms were monitored in real time via a Tektronix TDS5054B-NV oscilloscope and a high-voltage probe (LeCory PPE 5 kV Cusano Milanino, Italy). To maintain a fixed load of 50 Ω cells were exposed at a concentration of 6×105 cells in 100 μl of an artificial isotonic buffer, as fully detailed in [90], in which above 90% cell viability was assured up to over three hours of cell suspension in this buffer. Cells were irradiated using a Gilardoni CHF 320 G X-ray generator (Gilardoni, Mandello del Lario, Lecco, Italy) operating at 250 kVp, 15 mA, with HVL = 1.6 mm Cu (additional filtration of 2.0 mm Al and 0.5 mm Cu). The dose rate was 0.89 Gy/min at an irradiation distance of 67.7 cm.

### 4.3. Viability Assays

The Trypan blue exclusion test was used to determine the number of viable cells present in a cell suspension. Briefly, cell suspensions were mixed 1:1 with trypan blue dye and automatically cell viability was recorded with a LUNA-II™ cell counter (Logos Biosystem, Villeneuve-d’Ascq, France) 24 h after the exposures. Cell viability was also continuously monitored in real time by RealTime-Glo™ MT Cell Viability Assay (Promega, Milan, Italy), according to the manufacturer’s instructions. Luminescence was monitored at 0, 24 and 48 h after cell plating using a plate-reader GloMax (Promega, Milan, Italy).

### 4.4. Flow Cytometric Analyses

#### 4.4.1. Cell Permeabilization

Cell membrane permeability was analyzed immediately after the electric exposure (T = 0 h) or 3 h later (T = 3 h). Of note, prior to being transferred to the cuvette for PEF exposure, U87 NS were disgregated and reduced to a single cell suspension. A fluorescent probe (Yo-Pro-1, Life Technologies, Italy, λex = 490 nm, λem = 510 nm) at a concentration of 5 μM was present in the cuvette during the exposure of U87 ML and NS re-suspended in the electroporation buffer. Exposed cells (T = 0 h), were diluted in PBS and analyzed by flow cytometry (FACSCalibur®, BD Biosciences, Mountain View, CA, USA) within 10 min after the exposure. To check permeabilization at T = 3 h, exposed cells were plated and after 3 h stained and analyzed by the cytometer as aforementioned. Forward (FSC-H) and side scatterings (SSC-H) were employed to exclude cellular debris from the analysis and to gate the integer/healthy cells. Data were acquired by Cell Quest software and then analyzed by FCS Express v.7 (De Novo, Pasadena, CA, USA).

#### 4.4.2. Evaluation of ROS

ROS analysis was carried out in U87 ML and NS cells labeled by dihydroethidium (5 μM DHE, λex = 518 nm, λem = 605 nm) at 37 °C in the dark for 20 min, then the cells were washed and analyzed immediately after the electric exposure (T = 0 h) by low cytometry (FACSCalibur®, BD Biosciences, Mountain View, CA, USA). To check ROS at T = 3 h, exposed cells were plated and then labeled and analyzed by flow cytometry. Forward (FSC-H) and side scatterings (SSC-H) were employed to exclude cellular debris from the analysis and to gate the integer/healthy cells. The mean fluorescence intensity (MFI) was calculated for each sample as the ratio between the mean fluorescence values in the channel of the probe-labeled versus the sham-treated cells. Auto fluorescence of cells was subtracted using a negative sample.

#### 4.4.3. Cell Cycle

Briefly, about one million U87 ML and NS cells fixed in cold 70% ethanol solution were centrifuged, re-suspended in Propidium Iodide (PI)/RNase staining buffer (λex = 490 nm, λem = 600 nm) and kept in the dark at room temperature for 30 min. The cell cycle was analyzed 24 h after exposure by flow cytometry (FACSCalibur®, BD Biosciences, Mountain View, CA, USA). Samples were analyzed by flow cytometry as described above.

### 4.5. Transcriptomic Profiles of GBM Cells after PEF-5 Exposure

Both U87 ML and NS cells, 24 h after PEF-5 exposure, were subjected to whole transcriptome analysis through Clariom S Affymetrix chips (Affymetrix, Thermo Fisher Scientific, Waltham, MA, USA). In particular, in vitro transcription, hybridization and biotin labeling of RNA has been performed according to GeneChip™ WT Kit protocol and Clariom™ S human gene platform. Microarray data (CEL files) were generated using default Affymetrix microarray analysis parameters (Command Console Suite Software by Affymetrix). Then, CEL files were normalized using the robust multiarray averaging expression measure of Affy-R package (www.bioconductor.org 30 March 2021). Differentially expressed genes between sham- and PEF-exposed cells (N = 4 for each group) have been identified using the Significance Analysis of Microarray (SAM) algorithm coded in the samr R package [91]. Genes with a calculated FDR < 0.05 were considered significantly differentially expressed between the experimental groups. Expression data were deposited into the Gene Expression Omnibus (GEO) database under Series Accession Number GSE195506 and are accessible without restrictions. Gene lists were analyzed for their enrichment (FDR < 0.1) in pathways reported in multiple MSigDB gene sets including Hallmarks, and C2cp (common pathways). Enrichment maps in Figure 3e have been generated with enriched C2cp terms by using the Enrichment Map application in Cytoscape 3.9.0 software (https://cytoscape.org 18 October 2021).

### 4.6. RNA Isolation and RT-PCR

RNA isolation from cells was performed with an Rneasy Mini Kit (# 74104; QIAGEN, Milan, Italy). After quantification, 2μg of total RNA was reverse transcribed with a High-Capacity cDNA Reverse Transcription Kit (Applied Biosystems, Foster City, CA, USA), and qPCR was carried out as previously described [49]. Oligonucleotide primers used for quantitative RT-PCR are listed in the Appendix A. Reactions were performed in triplicate from each biological replicate. Relative gene expression was quantified using Glyceraldehyde-3-phosphate dehydrogenase (GADPH) as a housekeeping gene. The ΔΔCt quantitative method was used to normalize the expression of the reference gene and to calculate the relative expression levels of the target genes.

### 4.7. In Vitro Functional Testing

#### 4.7.1. Clonogenic Survival

The ability of the GBM cells to generate in vitro colonies after exposure was determined using the clonogenic assay, performed on both first neurospheres derived from U87 ML (U87 NS1) and second generation neurospheres derived from U87 NS cells (U87 NS2). Briefly, sham-exposed and exposed cells U87 ML and NS cells were plated at a concentration of 8.000 cells/well into 24-well containing two agar layers: 0.3% lower agar layer and 0.6% upper agar layer in which cells were resuspended. Cells were cultured in defined medium (DMEM/F12 supplemented with 0.6% glucose, 100 units/mL penicillin and 100 mg/mL plus 10 ng/mL of insulin, 2 mg/mL of heparin, 9.6 ng/mL of putrescine, 0.063 ng/mL of progesterone, hepes 5 nM, B27 1X without vitamin A, 20 ng/mL basic h-FGF and 20 ng/mL of h-EGF). Three hours later, cells were irradiated with different x-ray doses (0, 2, 5 and 8 Gy) and then incubated again for colony formation. The medium was regularly added three times a week, until colony formation (about three weeks). After 3 weeks, the colonies were fixed in 10% ethanol and stained with 0.1% crystal violet for 30 min at room temperature. Colonies that contained more than 50 cells were counted by captured images, and the survival was calculated as the average number of counted colonies divided by the number of plated cells, multiplied by plating efficiency (PE), where PE is the fraction of colonies counted and divided by cells plated without radiation. Surviving fractions were normalized by the plating efficiency of unirradiated sham controls.

#### 4.7.2. Matrigel Invasion Assay

Sham and exposed U87 ML and NS (1.25 × 10^5^ cells) in 200 μL of DMEM with 0.5% FBS were seeded into Matrigel matrix-coated upper 24 wells containing polyethylene terephthalate filters with 8 μm porosity (Corning^®^ BioCoat™ Matrigel^®^ Invasion Chamber; Corning, CA, USA). The lower chamber was filled with 10% FBS-containing medium. After 24 h, non-invading cells were removed from the upper surface of the membrane. Cells invading the Matrigel matrix and adhering to the bottom surface of the membrane were fixed and stained with 1:1 methanol + crystal violet. Using an inverted light microscope (Zeiss, Axio Observer, Jena, Germany), the number of invading cells was analyzed by Image J software measuring the area corresponding to the migrated cells in each picture of the insert’s membrane, as suggested in [92].

### 4.8. Statistical Analysis

Results are expressed as the mean of three biological replicates ± standard error of the mean (SEM). All statistical tests were performed with GraphPad Prism software v.7 (GraphPad, San Diego, CA, USA). *p* values were determined using a two-tailed *t* test. * *p* < 0.05; ** *p* < 0.01; *** *p* < 0.001; **** *p* < 0.0001.

## 5. Conclusions

Our dataset was obtained by analyzing different endpoints, i.e., cell viability, cell permeabilization, ROS generation, cell cycle perturbation and functional testing. The evidence suggests that PEF-5 treatment on homogeneous and more heterogeneous U87 GBM cell populations has a significant effect. PEF-5 exposure inhibited clonogenic and invasion potentials in both U87 ML and NS cells, reducing the ability to form new neurospheres and transmigrate in vitro. This exposure substantially influenced CSCs’ fate and specifically affected their proliferation by differentially regulating cell cycle checkpoints by ROS signal transduction processes. This physical stimulus is easily modifiable in terms of the number of pulses delivered and the degree of intensity and repetition frequency, and this technique could be adapted to target specific tumor cells to optimize personalized therapy alongside conventional oncologic therapies. PEF therefore represents a promising therapeutic approach to pretreat CSCs and cancer cells, reducing radio or chemotherapy doses and helping to prevent tumor relapse.

## Figures and Tables

**Figure 1 ijms-23-03001-f001:**
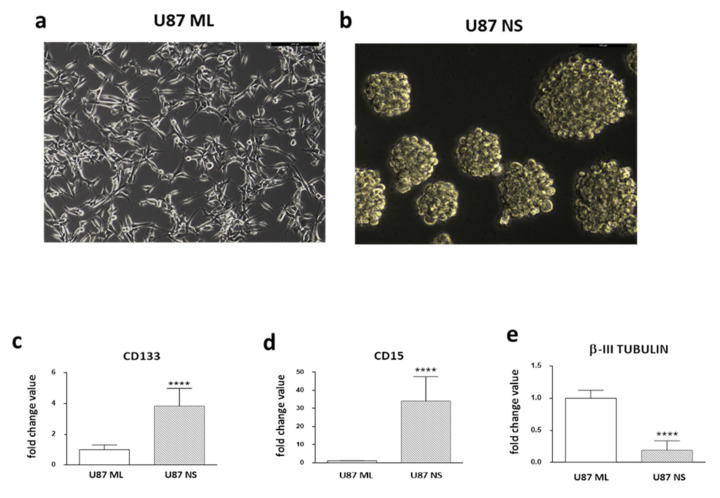
Analysis of stemness/differentiation markers in U87 ML and U87 NS. (**a**) Representative images of U87 cells grown in standard medium as a monolayer. (**b**) Neurospheres grown for 7 days in a selective medium. (**c**,**d**) *CD133* and *CD15* mRNA expression levels, considered stemness markers, and (**e**) *β-III TUBULIN*, a well-known marker of neuron differentiation in U87 ML and NS cells. Data are shown as relative expression compared to U87 ML. *p* values were determined using a two-tailed *t* test, **** *p* < 0.0001.

**Figure 2 ijms-23-03001-f002:**
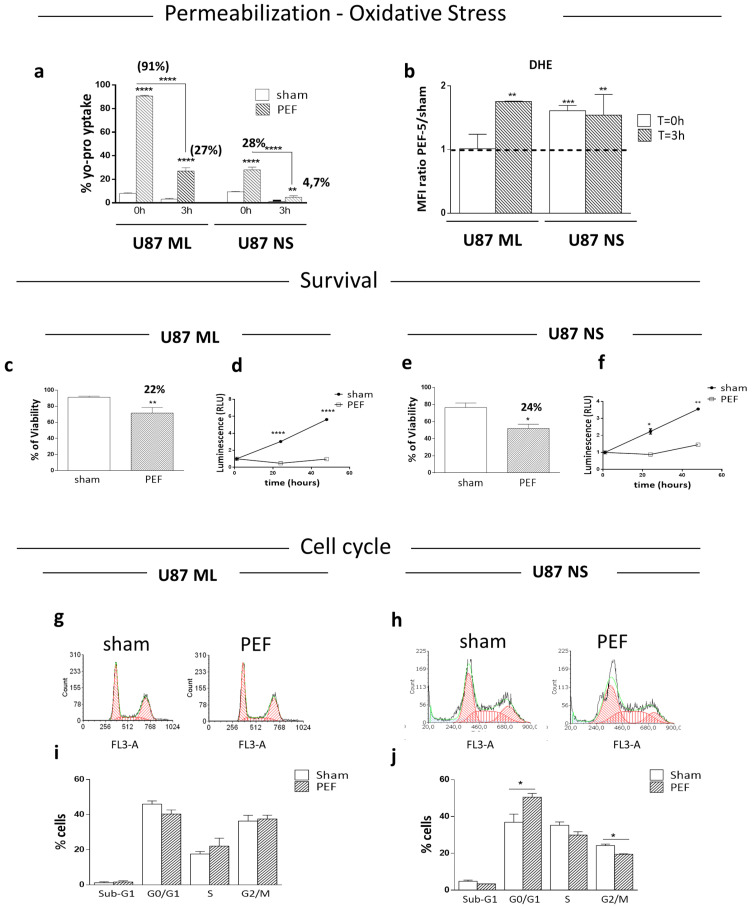
Membrane permeabilization, ROS production, cell survival and cell cycle. (**a**) Yo-pro-1 uptake at T = 0 h and at T = 3 h after PEF-5 exposure in U87 ML and U87 NS cells. (**b**) ROS levels after PEF-5 exposure at T = 0 h and T = 3 h. The histogram shows the ratio between the mean fluorescence intensity (MFI) of exposed and sham-exposed cells in both cell culture conditions. (**c**–**e**) Bar graphs of sham-exposed and exposed U87 ML and NS cells 24 h after PEF-5 exposure for cell death quantification (Trypan blue assay). (**d**–**f**) Cell proliferation evaluated at different time points (0, 24 and 48 h) after PEF-5 exposure in U87 ML and U87 NS. (**g**–**j**) Cell cycle phases in U87 ML and NS at 24 h after PEF-5 exposure. *p* values were determined using a two-tailed *t* test. * *p* < 0.05; ** *p* < 0.01; *** *p* < 0.001; **** *p* < 0.0001.

**Figure 3 ijms-23-03001-f003:**
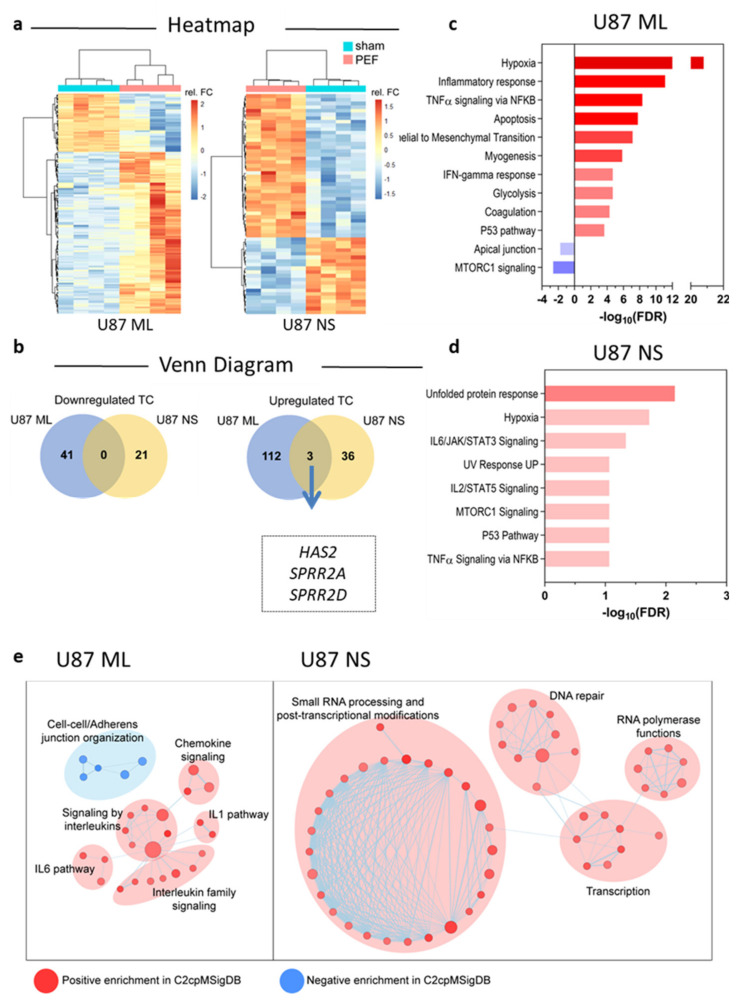
Transcriptomic analysis. (**a**) Heat maps displaying differentially expressed genes in U87 ML and NS cells. (**b**) Venn diagram depicting the number of significantly downregulated and upregulated genes in ML and NS U87 cells, as well as the name of commonly upregulated genes. (**c**,**d**) Bar graphs displaying the positive (red) and negative (blue) transcriptional enrichments in MSigDB gene sets of differentially expressed genes between sham-exposed and PEF5-exposed U87 ML and U87 NS cells. (**e**) Enrichment maps displaying transcriptional enrichments in cellular remodeling and inflammation processes in U87 ML cells and in DNA repair and RNA processing and post-transcriptional modification in U87 NS cells.

**Figure 4 ijms-23-03001-f004:**
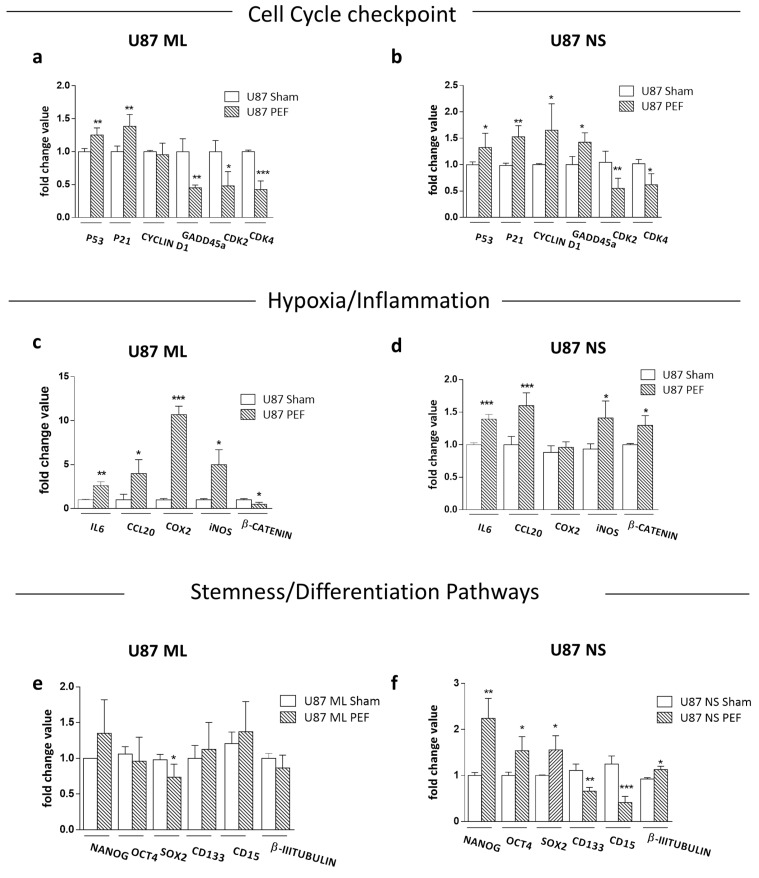
Cell cycle checkpoint, hypoxia/inflammation and stemness/differentiation markers. (**a**,**b**) Relative mRNA expression levels of genes associated with cell cycle, i.e., *P53*, *P21*, *CYCLIN D1*, *GADD45*, *CDK2* and *CDK4*. (**c**,**d**) Relative mRNA expression levels of genes associated with hypoxia/inflammation processes, i.e., *IL6*, *CCL20*, *COX2*, *iNOS* and *β-CATENIN*. (**e**,**f**) Relative mRNA expression levels of genes associated with stemness/differentiation processes, i.e., *NANOG*, *OCT4*, *SOX2*, *CD133*, *CD15* and *β-III TUBULIN*. *p* values were determined using a two-tailed *t* test. * *p* < 0.05; ** *p* < 0.01; *** *p* < 0.001.

**Figure 5 ijms-23-03001-f005:**
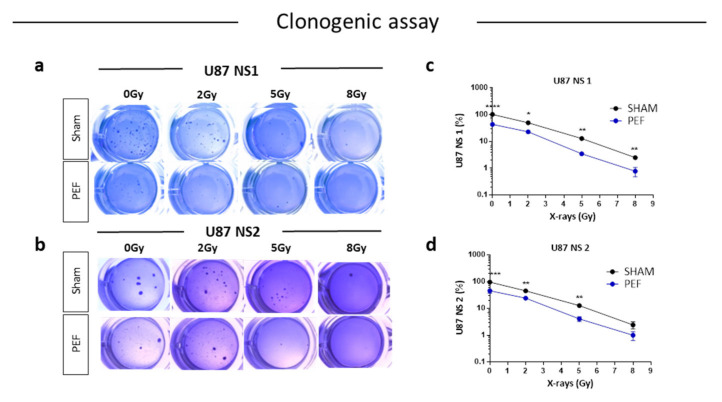
Survival clonogenic assay. (**a**,**b**) Representative images of primary and secondary neurospheres after PEF-5 exposure and IR treatments. (**c**,**d**) Quantitative analysis revealed a significant decrease in the clonogenic capacity between sham- and PEF-5 exposed cells, regardless of culture conditions. The combined treatment (PEF-5 + IR) reduced clone formation as a function of delivered radiation doses. *p* values were determined using a two-tailed *t* test. * *p* < 0.05; ** *p* < 0.01, **** *p* < 0.0001.

**Figure 6 ijms-23-03001-f006:**
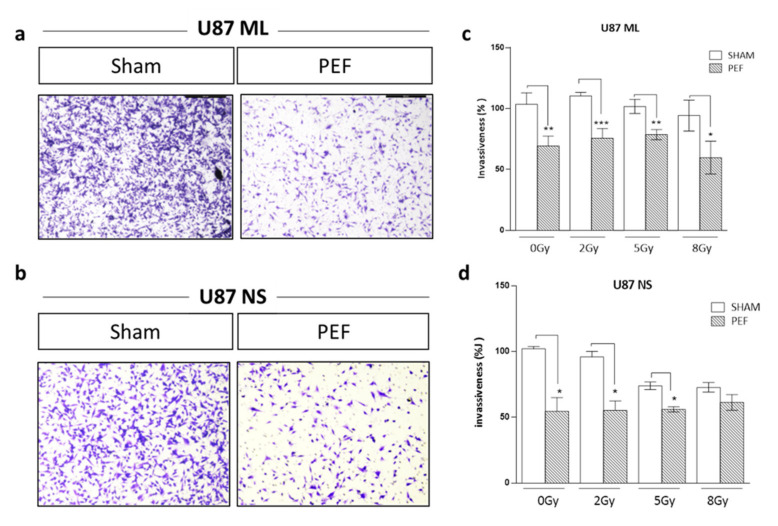
Invasion and migration assay. Representative images of transmigrated cells for (**a**) U87 ML and (**b**) U87 NS cells. (**c**,**d**) Percentage of invasiveness in U87 ML and NS cells evaluated 24 h after PEF-5 exposure and IR combined treatment. *p* values were determined using a two-tailed *t* test. * *p* < 0.05; ** *p* < 0.01; *** *p* < 0.001.

## Data Availability

All data are listed in tables or presented in figures in the main text or Appendix A. Moreover, gene expression data generated within this study have been deposited into GEO database under Series Accession Number GSE195506 and are accessible without restrictions.

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
