# Peer review of "Effects of Ultra-Short Pulsed Electric Field Exposure on Glioblastoma Cells"

_ijms, 2022, doi:10.3390/ijms23063001_

Round 1
Reviewer 1 Report
I’d like to thank the editors for the opportunity to review “Effects of Ultra-short Pulsed Electric Fields exposure on Glio-2 blastoma Cells” by Casciati et al.
The authors use U87 glioblastoma cells grown both as a monolayer and as neurospheres throughout a series of experiments looking at the effect of five electrical pulses (duration 40 µS, amplitude 0.3 MV/m) with an interpulse interval of 1 Hz, a protocol described as PEF-5.
Initially, the differentiation status of monolayer and neurosphere cultures was compared with monolayers showing lower levels of CD133 and CD15 transcript than neurosphere cultures, indicative of greater differentiation. Transcript levels of the neuronal differentiation marker β-III tubulin were also increased in monolayer cultures compared with neurosphere cultures. The manuscript describes PEF-5 leading to membrane permeabilisation in monolayer cultures to a greater degree than in neurospheres. PEF-5 treatment induced reactive oxygen species in monolayer but not neurosphere cultures (although the latter had higher pre-treatment levels which remained unaltered). PEF-5 treatment reduced viability in both neursophere and monolayer cultures and G0/G1 cell cycle arrest in neurosphere but not monolayer cultures. Large scale apoptosis was not seen in this analysis.
To understand the molecular mechanism of these observations the authors use transcriptomics to understand the potential signalling pathways and biological functions impacted by PEF-5 treatment. Both monolayer and neurosphere cultures show increased transcription of components of the hypoxia, p53 and TNFα signalling (via NFκB) pathways. Interferon γ signalling pathway components showed increased transcription in monolayer cultures and multiple JAK/STAT signalling components showed increased transcription in neurosphere cultures. However only three transcripts, those for HAS2, SPRR2A and SPRR2D were upregulated in both monolayer and neurosphere cultures.
The manuscript describes validation real time PCR experiments of selected genes differentially expressed following PFE-5 treatment identified by transcriptomics analysis. However the figure showing the results described in section 2.4.2 does not seem to be included in the manuscript or supplemental data.
The authors continue to describe the functional impacts of PEF-5 treatment on cultures. Both monolayer and neurosphere cultures showed reduced colony formation and invasion following PEF-5 treatment, and growth and invasion were not further impaired by irradiation treatment of cultures at 2 Gy, 5 Gy or 8 Gy 3 hours after PEF-5 treatment.
The authors conclude that pulse electrical field treatment has potential as an adjunct therapeutic approach to pre-treat tumours to prevent relapse.
The authors use appropriately controlled experiments involving well-established techniques to investigate the effects of PFE-5 treatment. Data are clearly presented and arguments logically developed. While beyond the scope of this manuscript, experiments expanding this analysis to incorporate normal cell populations (eg. in animal models that develop spontaneous tumours) will provide valuable information about the effect of treatment on normal tissues and the tumour tissue in the context of its microenvironment. Given upregulation of innate immune signalling pathways by 5-PFE the effect of this treatment on tumour immunogenicity should also be investigated at this stage.
I have one major concern that prevents me recommending publication of the manuscript – it must be revised to include real time PCR data (Figure 4) that appears accidentally omitted and is referenced in section 2.4.2.
While it does not prevent publication, protein-level confirmation of key aspects of the transcriptomics data would be valuable even if included as supplemental data. While the transcriptomics data and functional data presented are consistent with each other, protein level data may show subtle differences that provide important insights into unexpected future results.
Reviewer 2 Report
The authors show data that pulses electric field alter cell function and viability depending on U87 formation as neurospheres vs mono layer cultures
the reviewer would like to see validation studies in additional GBM cell lines, regardless of NS potential. This would validate PEF use more broadly inGBM clinical approaches (which the goal of any Jody of work related to GBM)
alternatively the authors should comment on PEF therapy on other culture modules that form spheroids in culture (ie mammary spheroids, or stem/iPS cells.
finally the radiation dose curve comparing PEF treatment of NS vs ML cells do not appear to be significantly altered. I think the authors should revisit this experiment, as the radiation dose curve is very clearly overlapping
Round 2
Reviewer 2 Report
The reviewer has observed the recent data and it clearly show PEF alone has a phenotype of impaired colony formation in ln the absence of radiation.
Again the authors should find a second cell line responsive to radiation that confirms the phenotypes reported after exposure to PEF
Author Response
Our comments are attached below.

Round 3
Reviewer 2 Report
The authors have satisfied reviewer comments